# Clinical Applications of Virtual Reality in Musculoskeletal Rehabilitation: A Scoping Review

**DOI:** 10.3390/healthcare11243178

**Published:** 2023-12-15

**Authors:** Elizabeth Chaplin, Christos Karatzios, Charles Benaim

**Affiliations:** 1Division of Physical Medicine and Rehabilitation, Orthopaedic Hospital, University Hospital of Lausanne, 1011 Lausanne, Switzerland; christos.karatzios@chuv.ch (C.K.); charles.benaim@chuv.ch (C.B.); 2Department of Medical Research, Clinique Romande de Réadaptation, Suvacare, 1950 Sion, Switzerland

**Keywords:** virtual reality, rehabilitation, musculoskeletal, orthopedics, disability

## Abstract

(1) Background: VR is based on a virtual world that creates sounds effects and videos that replace the real environment. Arising literature shows VR efficacy in the field of neurological rehabilitation (NR) and that its use is also taking a place in musculoskeletal rehabilitation (MSR), as a treatment of various disorders that cause disability and chronic pain. (2) Aim: We discuss the role of VR in MSR, presenting its use and development on acute and chronic musculoskeletal disorders, based on the most recent literature. (3) Design and Methods: Literature searches were conducted in the databases Pubmed and Medline up to 30 September 2023. The PRISMA-Scr Checklist was followed. (4) Results: A total of 51 records were included. The analysed studies were conducted within a variety of populations, musculoskeletal disorders, settings, and VR technologies. Only a few studies could statistically affirm the efficacy of VR in MSR, as mentioned for the rehabilitation of the upper limb. Nevertheless, the observed trend is an improvement of the global perceived effect compared to traditional rehabilitation. (5) Conclusion: VR allows for the personalisation of treatment with an adaptable treatment platform, which may improve the participation of the patient and increase acceptability and adherence to long-term rehabilitation programs. We provide recommendations and suggestions for future research and use of VR in musculoskeletal rehabilitation.

## 1. Introduction

Musculoskeletal rehabilitation (MSR) combines various therapeutic modalities in an interdisciplinary setting to help the individual to return to normal function or to improve his/her disabilities. Musculoskeletal disorders affect 20–30% of the world’s population, being the second leading cause of disability [1]. Rehabilitation can be structured in inpatient or outpatient settings, according to the severity and the type of diagnosis, deficiencies, handicap, and objectives. 

The rehabilitation program should ideally start early in the disease process to reduce the deleterious effects of inactivity and immobilisation. Goals are set at the beginning of the treatment program and revisited during follow-up(s) [2]. The treatment should focus on optimizing the healing process, restoring the biomechanical relations between the normal and injured tissue, and managing the pain and the disability. Occupational therapy is often added to regain independence in various activities depending on the patient’s profile. Rehabilitation programs are often planned for a medium- to long-term period. Usually, the more chronic the musculoskeletal disease is, the longer the treatment lasts. Chronic musculoskeletal disorders are typically characterised by chronic pain and functional disability [1]. Furthermore, patients suffering from such kinds of disorders are exposed to psychological stress and can develop behaviours characterized by kinesiophobia and avoidance [1]. In this scenario, it becomes more difficult for the patient to regularly follow a long-term rehabilitation program, and the risk of losing motivation becomes higher. To be effective, therapies should be graded, motivating, and related to real-life functional activities [3]. 

Ivan Sutherland first introduced the concept of virtual reality (VR) in 1960 [4], and it was initially implemented in the healthcare sector during the 1990s for the purpose of presenting complex medical data, particularly in the planning of surgical procedures.

VR is an innovative technology, consisting of a computer interface that creates a real-time simulation and interactions through auditory and visual sensorial systems [5]. It is based on a computer-based three-dimensional (3D) technology, providing a simulation of the real world. It allows people to live an enhanced digital experience [5]. VR can be non-immersive or immersive, augmented, or mixed, depending on the intensity of interaction with the virtual environment and the number of stimulated senses [5,6]. Non-immersive virtual reality (VR) entails the user viewing a computer-generated image of a virtual environment on a screen without complete immersion in the virtual world. On the other hand, immersive VR involves utilising a head-mounted display (HMD) unit to provide a multisensory experience for the user [7]. Among the utilities found in VR, there are the VR vox play, VR glasses with an inertial measurement unit (IMU), VR head-mounted displays (HMD), Kinect (coming from the English words kinetic and connect), and robotics. 

Supplementary tools like handheld devices, gloves, and vibrotactile platforms can furnish sensory feedback to users, allowing them to engage and interact with the virtual environment in the form of an avatar [7]. Wii Fit (Nintendo) is one of the most well-known applications used in VR [8]. Most of the recent literature concerning VR in rehabilitation focuses on NR, for patients with central nervous system lesions, with emphasis given on poststroke patients [9]. It is important to underline that the main difference between NR and MSR relies on the clinical goal of the recovery [9]. In NR, the central objective is to attain a cortical reorganization, potentially leading to the restoration of motor functionalities. Conversely, in MSR, the primary aims include restoring the functional range of motion (ROM), recovering muscular strength, and reducing pain by addressing psychological distress, kinesiophobia, and central sensitisation [10]. 

This scoping review aims to present the most recent literature about the application of VR in the field of MSR in order to clarify and summarize the last updates concerning its utility, and what should be pointed out to improve our knowledge and scientific baggage.

## 2. Materials and Methods

### 2.1. Design

The scoping review is the result of a review process targeting studies from 2016 until 2023 and conducted worldwide. A description of method is provided below. The review follows the PRISMA-Scr Guidelines. The review protocol is registered on Open Science Framework (OSF). 

### 2.2. Stage One: Identifying the Research Question

Three research questions were formulated to guide this scoping review: How has literature on VR and MSR evolved in the last few years?What sort of evidence exists in the field of VR and MSR?Which are the most important elements to consider in order to improve the understanding of current literature regarding VR and MSR?

### 2.3. Stage Two: Search Strategy

Pubmed and Medline were used as database libraries. References were handled using Zotero. We decided to analyse studies of Pubmed published since 2016 as our source literature. It is interesting to see on the graphic diagram of the publication years that the largest number of works were published between 2016 and nowadays (Figure 1). Our first search words were “virtual reality, rehabilitation”. As the aim of our study was to focus on MSR, we specify our research using the search terms “virtual reality, musculoskeletal rehabilitation”. Aiming not to miss related papers that do not appear when using ‘’virtual reality, musculoskeletal rehabilitation’’ as keywords, and to prevent any potential misinterpretations or inaccuracies in conclusions and suggestions, we extended our research.

This involved incorporating additional search terms by combining “virtual reality” with each of the respective domains:“virtual reality shoulder rehabilitation”;“virtual reality upper limb musculoskeletal rehabilitation”;“virtual reality complex regional pain syndrome rehabilitation”;“virtual reality lower limb musculoskeletal rehabilitation”;“virtual reality hip rehabilitation”;“virtual reality knee rehabilitation”;“virtual reality ankle rehabilitation”;“virtual reality neck pain rehabilitation”;“virtual reality low back pain rehabilitation”;“virtual reality amputation rehabilitation”.

### 2.4. Stage Three: Study Selection

A total of 4163 were identified with the first search words (“Virtual reality, rehabilitation”), from 1993 until 2023. Among these papers, 149 works were meta-analysis, 651 were clinical trials, 511 were randomized controlled trials, and 678 were reviews. Secondarily, with the keywords “virtual reality, musculoskeletal rehabilitation”, we found 118 papers since 1998 on Pubmed and 46 on Medline. Among the 118 on Pubmed, 21 were reviews, of which 15 were systematics reviews, 5 were meta-analysis, 17 were clinical trials, and 15 were randomized trials. Among the 46 found on Medline, we did not detect studies relevant to the use of VR for rehabilitation of musculoskeletal conditions. Finally, considering papers from 2016 to September 2023, we found 104 papers with the search words “virtual reality, musculoskeletal rehabilitation”. We included studies concerning the use of VR in musculoskeletal rehabilitation in adults. We excluded papers concerning future studies (study protocols and feasibility studies) and case reports. We also excluded papers discussing the use of VR for biomechanics analysis, digital health, implementation science, and those reporting the use of VR in other domains than MSR. We finally identified and included 22 studies, which are presented and discussed below. Table 1 shows the number of included studies according to the domain of MSR. Furthermore, among the papers found with the additional search words, we finally included 29 additional records. Figure 2 summarises the procedure. The total number of papers considered suitable for this review was 51. 

### 2.5. Stage Four: Charting the Data

At this stage, the first 22 studies were reviewed and presented by the first author. The second author reviewed the additional 29 studies, and they were not consistently presented individually. They were used to develop the arguments of VR in a specific MSK disorder and to strength the drawing of the conclusions. 

### 2.6. Stage Five: Collating, Summarizing and Reporting the Results

In this final stage, the outcomes of the review were summarised, discussed, and presented according the different MSK disorder. No formal methodological quality assessment of the included articles was conducted.

## 3. Results

### 3.1. Description of the Included Studies

#### 3.1.1. Shoulder Disorders

Brady et al. [7] explored physiotherapists’ beliefs on VR for shoulder rehabilitation. Sixteen physiotherapists had the opportunity to experience immersive and interactive VR (Oculus Quest) and play virtual sports, such as boxing. Significant improvements in ROM, strength, and function as assessed using the Constant–Murley score were found in all participants. Most participants believed that distraction accounted for the mechanism underlying the pain relief they experienced while using VR and suggested that VR offered an enjoyable way to exercise that could be motivational. Given the presence of a fall risk, particularly in unsupervised situations, concerns about safety arose concerning the use of immersive VR. VR was seen as an exciting opportunity, potentially opening new avenues for managing shoulder pain, but its very ’newness’ created some apprehension.

Belotti et al. [11] investigated the development of a telerehabilitation platform for shoulder motor function recovery based on the use of a Microsoft Azure Kinect device for both patients and medical personnel. Participants were requested to play games by performing the defined rehabilitation exercises both in a correct and in a wrong way to emulate the pathological condition. Results were promising since the platform allows for an objective evaluation of the patient’s gradual improvements.

Dejaco et al. [12] evaluated the validity and reliability of active ROM measurements of a VR system for people with and without shoulder pain. An Oculus Quest head mounted display loaded with a custom-built virtual goniometer was used. Smartphones were used as digital inclinometers for measurement. Results demonstrated systematic differences between ROM measurement values obtained with the two systems. Measurement results obtained with the current version of the VR-goniometer likely overestimated the ’true’ value of the patients’ shoulder flexion and caption ROM.

Specifying our research for VR and shoulder musculoskeletal disorder rehabilitation, we find three more studies valuable to mention. Alvarez et al. [13] showed with their experimental study that an embodiment-based immersive VR training program might be a useful therapeutic tool to help improve ROM in patients with movement-related shoulder pain in the short term. Pekyavas O. N. et al. [14] compared in their level I randomized control trial (RCT) the short-term effects of home exercise program and VR exergaming in patients with subacromial impingement syndrome. VR exergaming sessions (Wii) were found to be more effective than home exercise programs in the short term regarding functional outcomes; both groups presented a similar decrease in pain intensity. Si-Huei Lee et al. [15] discussed the effectiveness of a goal-directed shoulder exercise rehabilitation system combining interactive VR and wearable sensors, in a series of 16 patients with frozen shoulder.

#### 3.1.2. Upper Limb Disorder

Padilla-Castenada et al. [9] presented a robotic system used for the orthopaedic rehabilitation of the upper limb, including four modules (robotic rehabilitation device, VR games for motion task execution of the forearm, task difficulty adaptation module based on patient performance, therapist graphic interface). One experiment was conducted on healthy subjects and the second on patients undergoing rehabilitation sessions of the forearm. With this approach, they considered that three aspects were especially worthy of interest when comparing to the neurorehabilitation robotic systems: muscular strength, pain reduction, and the improvement of functional ROM. Results were construed as initial indications that the system was well received and that the proposed methodology is feasible.

Nawvi WN et al. [16] investigated in their pilot study the impact of gamification on the functional recuperation of patients with unilateral distal radius fractures (DRF). They included 20 patients who were randomly assigned to group A (gamification) and group B (conventional rehabilitation) in a 1:1 ratio. Patients in group A engaged in gaming activities using Oculus Quest head-mounted display (HMD), while those in group B followed a conventional rehabilitation program. Both groups participated in a 60-min daily rehabilitation program, five days a week, for a duration of four weeks. The results revealed significant improvements in pain, range of motion (ROM), grip strength, and functional independence in both groups. Notably, the gamification group demonstrated significantly greater enhancements in hand function and functional independence compared to the conventional physiotherapy rehabilitation group. The conclusion drawn from the study is that gamification seems to exert a substantial impact on the rehabilitation process following distal radius fractures (DRF).

By additional research and always concerning DRF rehabilitation, Matamala-Gomez et al. [17] showed that a rehabilitation program, based on developing ownership over a virtual arm and exercising it in immersive VR training, improves the functional ability with better ROM and lower disability of the fractured arm after the immobilization period compared to non-immersive VR and conventional training.

#### 3.1.3. Complex Regional Pain Syndrome (CRPS)

CRPS is a challenging disorder to manage in trauma rehabilitation, reason why we have decided to dedicate a special paragraph to this topic.

Solcà M et al. [18] conducted a crossover double-blind study with two groups of persons (one with CRPS and one from the healthy control group). Participants wore an Oculus VR Headset and saw a realistic three-dimensional virtual hand placed on a table in the VR environment in a heartbeat-enhanced VR (HEVR) system, integrating principles from immersive VR, mirror therapy, and data from multisensory body processing. Results supported Class III evidence of the efficiency of HEVR in reducing pain and improving motor limb function without need for tactile stimulation, avoiding the risk of allodynia and thus facilitating application of higher doses of therapy. 

A possible analgesic effect and functional improvement when using immersive VR rehabilitation for upper limb CRPS has been also shown in a pilot study with six participants [19]. A recently published scoping review showed an overall subjective and objective positive effect of VR in the management of CRPS, with most evidence toward pain reduction and body perception disturbances improvement [20]. 

#### 3.1.4. Lower Limb, Hip, Knee, and Ankle Disorder

Elaraby et al. [4] aimed to evaluate the effect of VR in ankle injury rehabilitation in their systematic review and metanalysis. VR programs, in comparison to traditional physiotherapy, demonstrated notable enhancements in gait parameters such as speed and cadence, muscle force, and perceived ankle instability. However, no significant difference was observed in the foot and ankle ability measure. Furthermore, the use of VR balance and strengthening programs resulted in significant improvements in static balance and perceived ankle instability. It is worth noting that the quality of most studies included in the analysis ranged from poor to fair.

Vogt S et al. [21] provided a comprehensive overview of VR technology employed for balance prevention and rehabilitation’s effects following musculoskeletal lower limb impairments. There were differences in intervention length, session frequency and duration, and balance outcome measurements in the two domains of VR balance training. None of the included studies indicated negative effects of VR balance training compared to traditional methods; none of them reported either any significant advantage when using VR for balance rehabilitation. Two studies indicated better effects of VR balance training in healthy adults. 

Ebrahimi et al. [22] studied the effects of VR in 26 young women with patellofemoral pain (PFP) through a brain mapping with the use of EEG. Participants were allocated to an intervention and control group for more than 6 months. The intervention involved providing lifestyle education along with 8 weeks of VR therapy, comprising 24 sessions, each lasting 40 min. In contrast, the control group solely received lifestyle education. The VR program consisted of Kinect Adventures and Kinect Sport games. The VR group also showed balance improvement as a significant improvement of secondary outcomes, including function, quality of life (QoL), pain, and brain mapping, compared to the control-group. Increases in alpha and theta power in the frontal, parietal, and occipital lobe were shown as results of VR use. An alpha increase in the VR group may indicate an improvement in concentration on their movements. According to their study, VR can be added to routine physical therapy interventions and can be considered as a holistic approach in the rehabilitation of PFP. 

Kiani S et al. [23] summarised the technical aspects of using VR for the rehabilitation of the lower limb after having included 20 studies on VR-based systems for people with various lower limb disorders, post-surgical or not. Nearly half of the studies used the Kinect sensor as a VR input device and the Unity game engine for visualisation. Other VR applications used were IMU technologies and HMD. According to their findings, the recent literature does not provide a detailed description of the technical aspects of the developed VR rehabilitation system of the lower limb, and therefore, it is not possible to re-use it with safety and knowledge.

Houston et al. [24] found that VR-based gait education (VR-GEd) in persons with lower limb musculoskeletal injuries at the onset of a 3-week, multidisciplinary, inpatient rehabilitation program—incorporating VR games tailored to the patients’ specific deficiencies—resulted in a greater reduction in pain interference. It also led to substantial improvements in general anxiety, although not to a clinically meaningful extent. VR-GEd did not have an impact on functional outcomes. However, patients gave high ratings to the sessions in terms of enjoyment and perceived value. They emphasized a clearer understanding of their injury and believed that the sessions could positively contribute to their recovery.

Additional research found articles worth mentioning. Two studies, one of which was a recently published RCT, showed the interest of VR use in rehabilitation after anterior cruciate ligament (ACL) reconstruction. The first study unveiled a more pronounced impact of immersive VR environments on knee biomechanics in individuals compared to the control group. This suggests that incorporating VR applications may be beneficial in rehabilitation programs aimed to target altered movement patterns [25]. The second showed the effectiveness of using immersive VR in addition to the conventional rehabilitation after ACL reconstruction, achieving better results in terms of pain reduction and subjective knee evaluation. No significant differences regarding limb loading, balance, ROM, and functional tests were found [26]. A VR training protocol has shown beneficial improvement compared to sensory-motor training in post-traumatic knee osteoarthritis (KOA) after ACL injury, concerning not only pain and function but also inflammatory biomarkers without effect on bone morphogenic proteins [27]. Two RCTs using VR training after total knee replacement (TKR) showed no superiority over conventional rehabilitation regarding improvement of outcomes with a possible better effect on global proprioception [28,29]. Similarly, VR home rehabilitation with wearable devices resulted in similar improvements in functional outcomes following total hip replacement (THR), but with a better global perceived effect compared to traditional rehabilitation [30].

### 3.2. Spine Disorders

#### 3.2.1. Neck Pain

Kragting et al. [31] aimed to investigate how pain-free ROM fluctuated when VR was used as visual feedback, either underestimating or overestimating the actual rotation, in individuals experiencing acute and chronic neck pain. A total of 71 participants wore a VR Headset and were submerged in a virtual forest, with three different locations. They were instructed to gradually turn their head to the left until pain began, then to the right until pain commenced, and finally to return it to the midline. It appeared that pain intensity significantly increased between the 6th and the 12th repetition. Their research showed no effect of visual feedback manipulation on pain-free ROM and no interaction effect between the visual feedback condition and the duration of pain. 

Orr et al. [32] wanted to assess the feasibility, safety, and suitability of exercise therapy utilising virtual reality (VR) in addressing neck pain disorders (NPD) and non-specific low back pain (NS-LBP), along with identifying appropriate outcome measures. Participants were equipped with the Pico Neo 2 (ByteDance) head-mounted display (HMD) and hand controllers. A total of 82 participants completed an average treatment duration of 127.1 days. A noteworthy improvement in the mean Modified Oswestry Low Back Pain Disability Index was observed, as reflected in a lower score indicative of an amelioration after the treatment. The second significant improvement concerned the Neck Disability Index. The study showed that VR treatment appears to be safe (no adverse events or side effects, no deterioration through treatment course reported by the participants). Significant reduction in disability was observed for NS-LBP, with a 17.8% improvement (*p* < 0.001), and for NPD, with a 23.2% improvement (*p* = 0.02), as measured by the Modified Oswestry Low Back Pain Disability Index and Neck Disability Index, respectively. While some outcome measures did not reach statistical significance, the data indicated a positive trend over time towards improved health. This suggests that VR should be taken into consideration in the management of both NS-LBP and NPD.

Among the studies identified through additional searches, noteworthy findings include the feasibility and safety of VR training within an interdisciplinary rehabilitation program for individuals with chronic neck pain [33,34]. This approach shows potential for improving various symptoms related to pain, depression, sleep and kinesiophobia. Additionally, it appears to enhance active cervical ROM, boost motivation for exercise, and offer valuable feedback [35]. At the same time, as indicated in a single-blinded RCT involving 44 patients, only the parameter kinesiophobia demonstrated a difference between VR and exercise after 3 months. The remaining outcome measures pertaining to pain, mobility, and beliefs were found to be similar [36].

#### 3.2.2. Low Back Pain

Groenweld TD et al. [37] conducted an RCT including 40 individuals and focused on the effect of VR application on the health-related QoL of patients with nonspecific LBP. VR sessions combined patient education explaining the maladaptive changes in the central nervous system with elements of several evidence-based cognitive-behavioural pain therapies. The design consisted of “travelling” through the nervous system, hearing a calm voice explaining the mechanisms of pain, with the possibility of five different games based on different psychological treatment principles (mindfulness, hypnotherapy and desensitization and reprocessing). Results show that 4 weeks of a self-administered behavioural therapy-based VR program for LBP do not seem to improve QoL but is well tolerated and may positively affect daily pain experience.

Thomas J.S. et al. [38] conducted a RCT employing a virtual dodgeball intervention specifically crafted to induce gradual increments in lumbar spine flexion. This approach aimed to diminish expectations of fear and harm by involving participants in an engaging and diverting competitive game. Participants of the game group completed 15 min of virtual dodgeball between baseline and follow-up, for three consecutive days. No significant effects were found on lumbar spine flexion, expected pain, or expected harm. However, virtual dodgeball was effective at increasing lumbar flexion within and across gameplay sessions. 

Liikkanen et al. [39] demonstrated the utilisation of data obtained from wearable devices as a digital metric for chronic low back pain (LBP) research. In their pilot clinical trial, the VIRPI trial, they evaluated a novel digital therapeutic intervention (DTxP) in adults with chronic LBP. Participants underwent approximately 30 daily treatment sessions using the Oculus Quest VR head-mounted device (HMD) and two handheld controllers (HHC), with 10 participants assigned to the standard care control group. This repeated exposure to movement in a trial setting provided an opportunity to examine movement data collected by VR devices and two wearables. The DTxP was a fully immersive VR experience with 24 modules over 6–8 weeks. Results suggested that participant movement improved over time, as for most of the participants the average velocity of all the sensors increased during these segments over the study. 

Nambi et al. [40] investigated the effects of physiotherapeutic exercise programs on imaging findings and inflammatory biomarkers in soccer players with chronic non-specific LBP. A total of 60 participants were divided into virtual reality exercise, isokinetic, and conventional exercise groups. Results showed that exercise with VR and isokinetic could decrease pain intensity, increase the muscle cross-sectional area and thickness, and positively alter inflammatory biomarkers compared with conventional training.

Additional research concerning LBP rehabilitation identified interesting results. VR treatment combined with exercise was more effective in reducing kinesiophobia than VR intervention alone [41]. Virtual walking integrated physiotherapy may improve pain, kinesiophobia, and function in the short term [42]. A multimodal neurorehabilitative strategy, utilizing augmented multisensory feedback to aid patients in regaining a correct body image and executing proper movements with painful body parts, showed promise in addressing the multidimensional aspects of pain [43]. Non-immersive VR distraction had a hypoalgesic effect during and immediately after the exercises, and so could be useful when pain reduction during exercise is the goal [44].

### 3.3. Amputation

Rehabilitation following limb amputation poses considerable challenges. Coordinated expertise from diverse specialists is essential to assist individuals in adapting to a new body perception and developing a revised mobility pattern. In addition, individuals experience specific phenomena, such as phantom limb pain (PLP).

Hao J. et al. [45] incorporated in their review 10 clinical studies, and the findings highlighted the favourable impact of VR on enhancing motor function in prosthesis training. This improvement encompassed areas such as balance, gait, and upper extremity outcomes. Participants reported a positive subjective experience, including enjoyment, during the VR intervention. Nevertheless, the question remained uncertain regarding whether VR could yield superior therapeutic outcomes compared to conventional rehabilitation. This uncertainty stemmed from the limited number of controlled studies and conflicting results reported.

Further research revealed that there is more extensive literature investigating the effects of VR on PLP compared to the literature focusing on VR’s impact on the mobility aspects of amputees. 

Prahm et al. [46] demonstrated that game-based interventions provide a useful addition to standard EMG training and can achieve better results in clinical outcome measures in rehabilitation of patients with traumatic transradial or transhumeral upper extremity amputation. Rami L. Abbas et al. [47] declared that adding VR training to a rehabilitation program in the late stages of prosthetic training could have superior effects on the balance outcomes but not on walking capacity in unilateral, traumatic lower limb amputees. Two systematic reviews reported a decrease in PLP scores following a single VR session or a VR intervention consisting of multiple sessions, an effect that was more significant when combining VR with tactile stimulation. It is interesting to consider that VR stimulation can active different cerebral regions, leading to a reduction of the maladaptive cortical reorganization associated with PLP [48,49].

### 3.4. General Musculoskeletal Disorder and Chronic Pain

Our research found quite a few articles, especially literature reviews, on the use of VR in rehabilitation of various musculoskeletal conditions. 

The use of interactive VR may be suggested to decrease overall pain compared to scenarios with no rehabilitation and traditional rehabilitation. However, it does not confer additional benefits in terms of functional disability, as indicated in a review encompassing studies involving patients with chronic MSD (ankylosing spondylitis, fibromyalgia, lateral ankle sprain, neck, and low back pain). Non-immersive iVR showed improvement in psychological distress as compared with no rehabilitation. However, no statistically significant difference in the outcomes existed between non-immersive and immersive iVR [1]. Evidence of VR effectiveness was described as promising in chronic neck pain and shoulder impingement syndrome. VR and exercises have similar effects in rheumatoid arthritis, knee arthritis, ankle instability, and post-anterior cruciate reconstruction. For fibromyalgia and back pain, as well as after knee arthroplasty, the evidence of VR effectiveness compared with exercise is absent or inconclusive [6].

Schuermans J et al. [8] found in their systematic review that VR could provide a benefit for both sports injury prevention and rehabilitation outcomes by offering clinicians with the opportunity to address the underlying biomechanical risk profile for common sports injuries, allowing the athletes to train protective movement patterns more effectively.

VR might be useful to reduce musculoskeletal pain even if no conclusive clinical recommendations can be made based on the low quality of available research [50,51], while gaming was found not superior to other treatments for reducing pain catastrophising, anxiety, depression, and pain-related fear (findings based on very low- or low-quality evidence) [52].

VR could improve treatment motivation by effectively distracting patients with chronic pain, allowing them to ignore the cumbersome rehabilitation training with—in addition—a possible positive psychological effect, as concluded by Lin HT et al. in their review exploring the VR treatment effects for all musculoskeletal disorders [53]. VR appears also to have a significant effect upon joint mobility or motor functions of patients with chronic painful musculoskeletal disorders.

## 4. Discussion

The potential of VR’s use in shoulder pathology seems favourable, and adding VR to existing rehabilitation modalities could eventually provide benefit by reducing workload of rehabilitation specialists and the cost of hospitalisation rates and readmissions. It could be used as telerehabilitation in a remote setting by improving progress monitoring, providing real-time feedback, and improving adherence to exercise programs, considering that VR is user-friendly if used safely, could improve patients’ engagement to exercise programmes, and seems to help in terms of functional outcomes. Two different VR systems were used (Oculus Quest in the first study and Kinect for the second). As described by Tokgoz et al. [54], VR technologies have the potential to become an effective tool in the functional rehabilitation process of patients with injuries and diseases of the upper extremity. The first study presented a robotic system for the use of VR, and the second study used the head-mounted display, showing both promising results. For shoulder and upper limb disorders, we suggest the combination of robotics and HMD with supervised sessions, to follow the improvement of the ROM and reduce the risk of self-injuries. In our opinion, VR is worth using in CRPS’ rehabilitation, as any modality aiming to reintegrate the affected limb in the body schema and image, based on cortical reorganisation and pain modulation pathways. 

VR is promising in lower limb musculoskeletal rehabilitation and it seems that Kinect results to be generally the most used device, followed by IMU technologies and HMD. The findings are interesting in knee osteoarthritis (OA) and PFP, and positive were the results in individuals after ACL reconstruction, but not in those following TKR or THR. This difference might reflect the role of CNS in proprioception after tendon/ligamentous injuries. As Lee M. et al. [55] concluded, it could be useful to use various levels of difficulty to meet patients’ needs and by considering the knee injury’s severity. Considering that VR-based games are potentially acceptable as a motivational rehabilitation tool for patients following knee surgery [56], in our opinion, the use of VR in the rehabilitation of patients with lower limb disorders should be encouraged as it could be used as an alternative to conventional rehabilitation, with the possibility of a longer treatment period and more frequent sessions, even if results seem not always necessarily superior to the standard approaches. 

According to the selected studies on neck and LBP, HMD was the most used application, with studies reporting a global improvement of the perception of disability for both neck and low back pain and a positive effect in the daily pain experience. In our opinion, is important to consider the subjective perception of the patient as a relevant factor in considering the utility of VR therapy, as estimated in the first two studies on LBP which focus their attention on using the VR therapy as entertaining and distracting, to achieve pain desensitisation. VR can induce an external focus, which can be a useful and promising technique for defocusing, fighting barriers such as a lack of motivation and negative attitudes towards chronic pain. We suggest extending the duration of the studies in order to analyse the results after long-term use of VR. Finally, we agree with the opinions expressed in the two systematic reviews reporting that VR, combined with the interdisciplinary approach, can be promising to reduce pain and kinesiophobia in patients with chronic LBP and improve pain disability and mobility in patients with chronic NP. Nevertheless, high heterogeneity, lack of VR intervention design consistency (objective outcome measures, follow-up reporting, and large sample sizes) make it difficult to propose specific protocols [57,58].

From our point of view, more systematically integrating VR in post-amputation rehabilitation would be of special interest given its specificities compared to the rehabilitation of individuals suffering of other musculoskeletal disorders, as it implies the coordination of the central and peripheral neuromuscular mechanisms and the embodiment of prosthetic limbs, showing itself to be an effective treatment not only for PLP but also for prosthetic mobility. The use of VR in extended periods should be tested for chronic conditions such as CRPS and PLP, and maybe for their prevention.

Among the articles talking about general musculoskeletal disorders, we found an overall positive prospective in the use of VR in their rehabilitation. Quite interesting have been findings of the positive effects of VR on inflammatory biomarkers and on changes of brain functioning, as well as those concerning the importance of limb embodiment to obtain pain-related outcome improvement. However, the clinical heterogeneity of the disorders compared in the same study and of the participants can lead to bias and influence the conclusion; some papers were based on research performed on healthy individuals, and therefore were not able to observe a proper therapeutic effect in individuals suffering of an MSD. Another relevant aspect consists of the secondary effects of VR, as some studies observed motion sickness and dizziness with a risk of falling, leading them to consider these effects as a primary element to investigate before starting the VR treatment, according to the individual profile and comorbidities. Generally, current limitations to the use of VR technology are lack of awareness, the high cost associated with VR devices, and the lack of VR experiences tailored towards patients. 

Our study has limitations. Firstly, it is not a systematic review. Secondly, only one database was used as a search platform. No risk-of-bias assessment of included studies was enforced. We aimed to cover recent literature on the effect of VR on the rehabilitation of individuals with the most frequent and disabling musculoskeletal pathologies. Literature is vast, and to draw firm conclusions on the utility of VR in MSR, there is a need of supplementary studies on the use of VR specific to each MSK condition. The heterogeneity of methods and outcome measures contribute also to a difficult interpretation of the results, and the quality of the evidence needs to be higher. 

## 5. Conclusions

According to our analysed papers, only a few studies could statistically affirm the benefit of VR in MSR. The heterogeneity of the type of VR system in studies blocks the possibility to draw firm conclusions. Even though head-mounted displays, Kinect, robotics, and joysticks seemed to be the most used, the duration of the sessions provided as well as the virtual setting in which the individuals were put, could lead to different results. Nevertheless, the observed trend was that this new typology of treatment has great potential and is appreciated by both patients and clinicians. VR treatment appears to be promising to allow the patient distraction, leading to an increase in motivation, which could improve the individual’s compliance to long-term rehabilitation programs. 

## 6. Future Directions

In order to give a consistent contribution in the development of MSR, we suggest extending the studies with a more large-scale prospective research with clinical trials, blinded assessor, and intention-to-treat analysis. The designs of VR systems and the criteria for measuring outcomes should be more consistent to yield substantial evidence regarding the effectiveness of various VR types, whether used independently or in conjunction with other therapeutic approaches. Its application in MSR requires meticulous planning and feasibility testing to determine individuals who could derive the greatest benefits from its utilisation, or, conversely, might be adversely affected. A stronger collaboration between health professionals and VR developers could improve the quality of the rehabilitation programs. VR will probably never replace human contact with members of an interdisciplinary rehabilitation team but offers possibilities to reinforce the already-positive results of conventional MSR.

## Figures and Tables

**Figure 1 healthcare-11-03178-f001:**
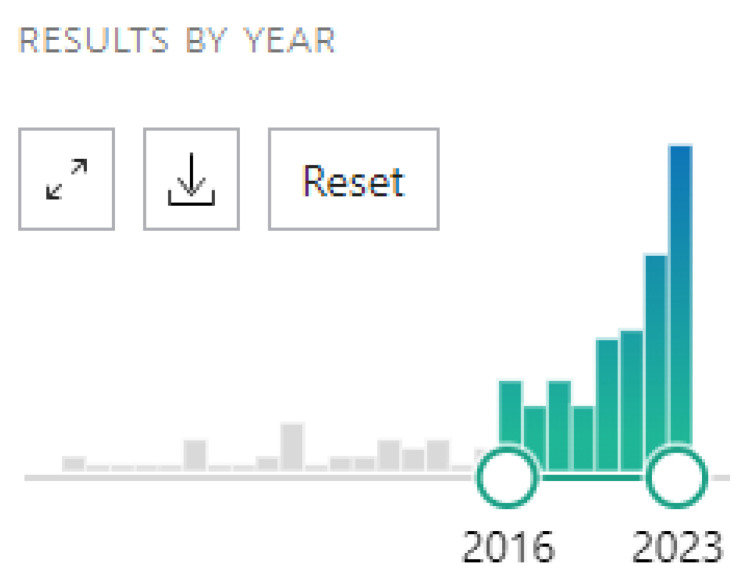
Diagram of the publication rate on PubMed (1998–2023).

**Figure 2 healthcare-11-03178-f002:**
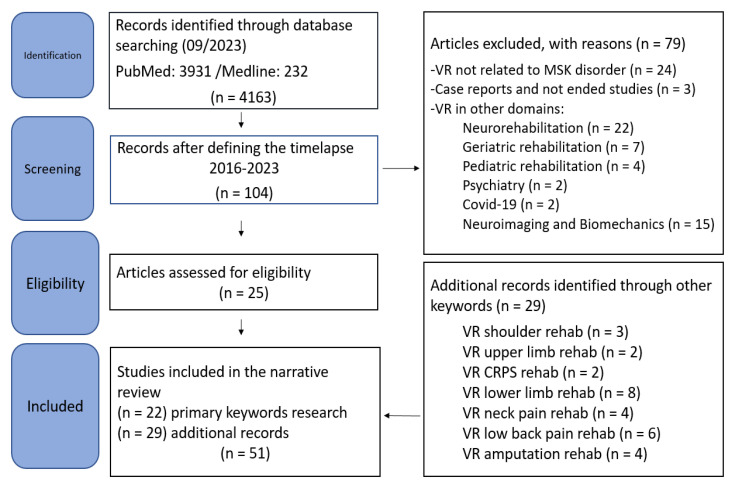
Flowchart of the studies.

**Table 1 healthcare-11-03178-t001:** Musculoskeletal disorders exposed in our paper’s selection (January 2016–September 2023).

Domains of MSR Discussed	Number of Studies	First Publication	Last Year of Publication
Shoulder	3	2017	2023
Upper limb	2	2018	2023
Complex regional pain syndrome	1	2018	-
Lower limb (hip, knee, ankle)	5	2019	2023
Spine disorders (neck, low back)	6	2017	2023
Limb amputation	1	2023	-
General musculoskeletal pathology and pain	4	2017	2023

## Data Availability

No new data were created or analysed in this study. Data sharing is not applicable to this article.

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
