# Peer review of "Clinical Applications of Virtual Reality in Musculoskeletal Rehabilitation: A Scoping Review"

_healthcare, 2023, doi:10.3390/healthcare11243178_

Round 1
Reviewer 1 Report
Comments and Suggestions for Authors
ABSTRACT
1. "Literature searches were conducted in the databases Pubmed and Medline from June 2023 until September 24 2023. " --This sentence is misleading to the reader are you saying that articles only from june to september 2023 are included. Clearly, the search based on your methods was from 2016-2023. I would revise this sentence and just write the month, date, and year of the final date you searched PubMed not a range.
2. The results section of your abstract is convoluted and meaningless. You are not providing any information of the results. Put only 1 sentence on the search results i.e. the number of records in the analysis. the next few sentences you should very succintly summarize the main findings of the results for the reader.
3. Consider shortening the background and aims of the abstract to put more information in the results which is the key point of the review.
INTRODUCTION
1. Define MSR at its first usage in the main text.
METHODS
1. Effectively only one database was used in this scoping review. MEDLINE is a part of PubMed. Ideally the authors should have included other databases as there are many journal articles not indexed in PubMed. The authors can either include these sources i.e. ScienceDirect, Scopus, EMBASE, etc. or open-access databases such as Hinari/Research4Life can also be considered. If the authors are not willing to include other databases into the scoping review I reccommend it clearly be stated in the discussion and conclusion as a limitation.
2. The review should following PRISMA-Scr guidelines not PRISMA. This is the adapted guidelines for scoping reviews.
3. Although it is not a systematic review, the review protocol should be registered. Open Science Framework (OSF) is a good option.
DISCUSSION
1. A formal paragraph on future directions proposed by the authors should be provided based on the findings of the scoping review.
2. Another limitation to the study is inherent to the scoping reviews that is no risk of bias assessment or quality assessment of included studies are enforced. This should also be included to inform the reader.
Comments on the Quality of English LanguageOnly minor editing for English language needed. Some sentences are compound and convoluted try shortening them. In general try to avoid superfluous sentences that do not add meaning to the manuscript. There are paragraphs in the discussion that are only 1-2 sentences. It is best practice that a paragraph be at minimum 4 sentences. As this makes your manuscript appear novice for structure. Consider joining paragraphs together.
Reviewer 2 Report
Comments and Suggestions for Authors
Thank you for conducting this paper.
I found the title of the paper to be interesting, and the content was well-written. However, I have some concerns about the methodology used in the paper. Typically, authors write narrative reviews or updates when there have been some significant developments in the related field. Additionally, meta-analysis usually covers the latest updates in the related field. Therefore, I am curious why the authors of this paper limited their search strategy to papers published only since 2016 when there are newer papers available. Also, I'm wondering how many authors reviewed the included articles. Was it double-checked, or did only one author select the papers?
Sincerely
Round 2
Reviewer 2 Report
Comments and Suggestions for Authors
Dear authors,
Thank you for your general major revision. All the concerns were addressed well. Now, it is ready for publication.
Sincerely